# Resistance to *Mycobacterium tuberculosis* infection among highly TB exposed South African gold miners

Violet N. Chihota[1,2,3]*, Thobani Ntshiqa[1], Pholo Maenetje[1], Raoul Mansukhani[4], Kavindhran Velen[1], Thomas R. Hawn[5], Robert Wallis[1,3], Alison D. Grant[2,4,6], Gavin J. Churchyard[1,2‡], Katherine Fielding[2,4‡]

1 The Aurum Institute, Johannesburg, South Africa, 2 School of Public Health, University of the Witwatersrand, Johannesburg, South Africa, 3 Division of Infectious Diseases, Department of Medicine, Vanderbilt University School of Medicine, Nashville, Tennessee, United States of America, 4 TB Centre, London School of Hygiene & Tropical Medicine, London, United Kingdom, 5 Department of Medicine, University of Washington, Seattle, Washington, United States of America, 6 Africa Health Research Institute, School of Laboratory and Medical Sciences, College of Health Sciences, University of KwaZulu Natal, Durban, South Africa

‡ These authors are joint senior authors on this work.
* vchihota@auruminstitute.org

**Data Availability Statement:** The data are provided on the London School of Hygiene and Tropical Medicine (LSHTM) Data Compass repository

## Abstract

### Background

Despite high exposure to *Mycobacterium tuberculosis*, a small proportion of South African goldminers resist TB infection. We determined, among long-service gold miners i) the proportion who were TB uninfected and ii) epidemiological factors associated with being uninfected.

### Methods

We enrolled HIV-negative gold miners aged 33–60 years with ≥15 years' service and no history of TB or silicosis. Miners were defined as TB uninfected if i) QuantiFERON-TB Gold Plus (QFT-Plus) negative or ii) in a stricter definition, QFT-Plus-negative and zero-response on TST and as resisters if they were of Black/African ethnicity and negative on both tests. Logistic regression was used to identify epidemiological factors associated with being TB uninfected.

### Results

Of 307 participants with a QFT-Plus result, median age was 48 years (interquartile range [IQR] 44–53), median time working underground was 24 years (IQR 18–28), 303 (99%) were male and 91 (30%) were QFT-Plus-negative. The odds of being TB uninfected was 52% lower for unskilled workers (adjusted odds ratio [aOR] 0.48; 95% confidence interval [CI] 0.27–0.85; p = 0.013). Among 281 participants of Black/African ethnicity, 71 (25%) were QFT-Plus negative. Miners with a BMI ≥30 were less likely to be TB uninfected (OR 0.38; 95% CI 0.18–0.80). Using the stricter definition, 44.3% (136/307) of all miners were

through request – please see https://doi.org/10.
17037/DATA.00002424.

**Funding:** G.J. Churchyard received funding from
The Bill and Melinda Gates Foundation Grant
number: OPP1116635 Full name of funder: Bill and
Melinda Gates Foundation Funder URL: https://
www.gatesfoundation.org/ The funders had no role
in study design, data collection and analysis,
decision to publish or preparation manuscript.

**Competing interests:** The authors have declared
that no competing interests exist.

classified as either TB uninfected (35; 26%) or infected, (101; 74%) and the associations
remained similar. Among Black/African miners; 123 were classified as either TB uninfected
(23; 19%) or infected (100; 81%) using the stricter definition. No epidemiological factors for
being TB uninfected were identified.

## Conclusions

Despite high cumulative exposure, a small proportion of miners appear to be resistant to TB
infection and are without distinguishing epidemiological characteristics.

## Introduction

*Mycobacterium tuberculosis* (*Mtb*) remains a leading infectious cause of mortality globally [1].
Just under a quarter of the global population was estimated to be latently infected with *Mtb*
[2]. Some individuals resist infection through innate immune responses [3]. Without treat-
ment about 5–10% of TB infected individuals, progress to active disease at some time in their
lives, half of whom develop tuberculosis (TB) within a few years after exposure [4]. This risk of
progression to active TB increases in HIV positive individuals in particular those with low
CD4 counts [5]. In South Africa, one of the high burden countries, TB incidence is highest in
goldminers where high prevalence of HIV, exposure to silica dust, and congregate working,
living, and social conditions contribute to ongoing transmission [6]. The gold mines have
been recognized as hotspots for TB with high tuberculosis risk estimated at 3% per annum [6,
7]. Mathematical modeling suggests that the annual risk of infection in this setting is as high as
20% which is five times more than the general population [8].

The risk of becoming infected with *Mtb* increases with repeated exposure and the infec-
tiousness of a source case. Though no direct measure for *Mtb* infection exists, a positive tuber-
culin skin test (TST) and/or interferon gamma release assays (IGRAs), such as QuantiFERON
Test (QFT) are used as surrogates for *Mtb* infection. Despite repeated exposure to *Mtb* in TB
endemic settings, evidence from longitudinal cohorts of household contacts of persons with
active pulmonary TB disease, in Uganda, suggest that a small proportion of individuals who
are repeatedly exposed to *Mtb* remain persistently negative on TST and IGRA [9, 10] and are
thought to resist *Mtb* infection [11], often referred to as "resisters". However, in studies of
household contacts, no epidemiological factors have been identified that distinguish individu-
als who resist *Mtb* infection from those who do not among adults [12–14].

Gold miners in South Africa have a greater risk of *Mtb* infection and disease, than the gen-
eral population. However resistance to TB infection in this population is poorly understood. A
recent cross-sectional survey conducted among gold miners in South Africa, estimated the
prevalence of TB infection, as measured by TST, to be very high, at 89%. Despite working in
this setting with high force of infection, an unanticipated small proportion of HIV negative
miners (13%) had a negative TST (zero induration) [15], suggesting no evidence of *Mtb* infec-
tion. This study did not adequately assess variables that may be associated with TB exposure.
Participants were included in the study irrespective of TB exposure status. The study was not
designed to restrict to individuals who were highly exposed to TB nor did it specifically charac-
terise the resister phenotype with respect to TST and IGRA. Mathematical modelling suggests
that long serving gold miners without evidence of *Mtb* infection have a very high likelihood
(93%) of having a resister phenotype [16].

Similar to household contacts, it is conceivable that a small group of miners may also resist *Mtb* infection. However no studies have been done to characterise individuals that resist infection among those highly exposed to *Mtb* in this setting. We hypothesized that some individuals that have been exposed to *Mtb* do not get infected and that it will be possible to identify highly TB exposed, uninfected miners and identify any epidemiological factors associated with that phenotype versus being TB infected (TST ≥5mm and QFT positive). Using a cross-sectional study design, we set to identify miners that remained TB uninfected (resist *Mtb* infection) despite being highly exposed to *Mtb* in the Highly Exposed TB Uninfected (HETU) study. We recognised that for the HETU study we would need to screen a large number of individuals to identify these individuals. The aim of this analysis was to explore factors associated with being TB uninfected in this highly exposed population.

## Materials and methods

### Ethics statement

The ethics committees of the University of Witwatersrand and London School of Hygiene & Tropical Medicine approved the HETU study. All consenting participants gave written informed consent, or for illiterate participants, witnessed verbal consent from an impartial witness.

### Study population and sites

The HETU study was conducted among HIV-negative gold miners attending the occupational health centre (OHC) for annual medical examination in North West Province, South Africa.

### Study procedures

Eligibility screening and enrolment. To identify individuals who had high cumulative exposure to TB and yet remain resistant to TB infection we screened potential participants for eligibility in two phases as described below.

**Pre-screening phase.** Goldminers, attending the OHC for their annual medical examination between August 2015 and December 2016 were pre-screened to exclude individuals who did not have prolonged exposure to TB based on proxies of age and years in the mining workforce (age <33 years and worked in the mines for <15 years). Age and years working in the mining industry were ascertained while miners were waiting to see a doctor for their annual medical examination. Individuals identified as having worked for prolonged periods in the mining industry (at least 15 years) and who were at least 33 years of age were consented and underwent full screening.

**Full screening phase.** A questionnaire was administered to collect information on medical history, including, prior and current treatment for TB disease and infection and a symptom screen for TB using the World Health Organization (WHO) screening tool comprising any of current cough, fever, night sweats or weight loss. The most recent chest radiograph taken as part of routine annual examination at the Occupational Health Centre were reviewed for evidence of old or active TB and silicosis. Participants were offered an HIV test, with pre and post-test counselling and an Oraquick HIV test was done by study staff, with referral to the mine health service for HIV treatment and care where needed. The full screen was done in five stages (Fig 1), whereby an individual who did not satisfy an earlier stage was excluded. The main purpose of this full screen was to exclude those who either had a past history or current TB, or those at higher risk of TB disease using markers such as TB preventive treatment, any serious or chronic medical condition, silicosis, being HIV positive and BMI <18.5. Of those satisfying both screening phases (high cumulative exposure to Mtb based on proxies of age

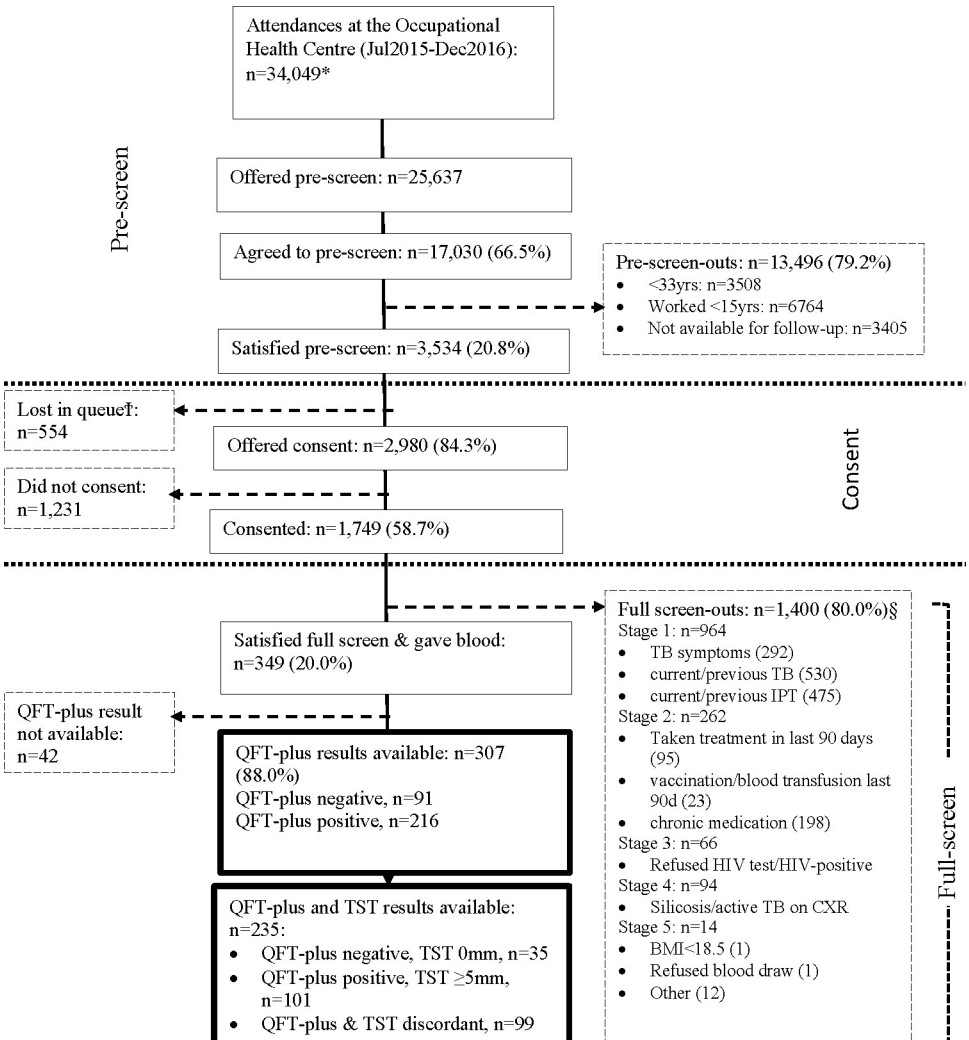

**Fig 1. Participant flow.** * Data from the daily attendance register. Includes repeat attendances during this time period. §Screen out questions were asked in stages; if screened out at a stage no screening was conducted for subsequent stages. Within a stage more than one screen-out reason may apply. yrs years; IPT isoniazid preventive therapy; BMI body mass index, QFT-plus QuantiFERON-TB Gold-Plus, TST tuberculin skin test. Lost in queue–miners who prescreened for the HETU study while waiting in queue for annual medical examination, but did not make it to the research site for further study related procedures.

and years in the workforce; no TB disease; or not being at higher risk of TB disease), we then assessed for TB infection using TST and QuantiFERON-TB Gold Plus (QFT-Plus).

Evaluation of *Mtb* infection using QuantiFERON-TB Gold Plus and Tuberculin skin test. Participants had a questionnaire administered to collect demographic characteristics, clinical and epidemiological data. A blood sample was collected for QuantiFERON-TB Gold Plus (QFT-Plus; Qiagen, Hilden Germany), interpreted according to manufacturer's instructions [17] and a spot sputum sample collected for mycobacterial culture on BACTEC MGIT 960 system (BD Diagnostic Systems, Sparks, MD, USA), to exclude culture positive TB. The QFT-Plus tests were done at the Aurum Institute, Clinical Research Laboratory, Klerksdorp and sputum cultures were done at BARC laboratories in Johannesburg. The TST was administered by trained research nurses using the Mantoux method, which uses five tuberculin units of purified protein derivative (Statens Serum Institute, Copenhagen, Denmark) and read, ideally,

after 48–72 hours and within 7 days. The TST response was measured using a digital calliper as the maximum transverse diameter of the induration expressed in millimetres.

During the first four months of the study, (10 July- 29 October 2015) under the first version of the protocol, participants had a blood sample collected for QFT-Plus at enrolment and a second blood sample collected for QFT-Plus and TST placed 90 days after enrolment. The protocol was subsequently simplified to collect a blood sample for QFT-Plus at enrolment and place the TST at enrolment or seven days later. Participants with a negative QFT-plus result at enrolment, regardless of protocol were followed up to 12 months where QFT-Plus and TST were repeated. The 12-month follow up data were not used to define the phenotype.

The HETU study was designed to identify at least 50 highly exposed TB uninfected participants. In order to identify epidemiological factors associated with being TB uninfected, the planned sample size was 100 QFT negative and 669 QFT positive miners at baseline.

Study outcome and definitions. The main outcome was the proportion of participants who were TB uninfected, defined as being negative on QFT-Plus, at enrolment regardless of protocol version, so as to maximize the sample size. The QFT-Plus test was considered negative if both antigen tubes (TB1 or TB2) tested negative as per manufacturer's recommended cut-off (an antigen response < 0.35 IU ml above the negative control response).

For purposes of a sensitivity analysis, using a stricter definition, we included participants with QFT-Plus and TST measurements within seven days of each other: either 90 days after enrolment for participants enrolled under the first version of the protocol or at baseline for participants enrolled under the second version of the protocol. The stricter definition for being TB uninfected (subsequently referred to as being a resister), was being QFT negative and having a zero TST response (TST = 0mm), and TB infected as QFT-Plus positive and TST positive (induration ≥5mm). Miners with discordant results were excluded from the analysis. By further restricting the analyses to Black/African miners, we excluded all miners from other race groups that would have been TST/QFT-Plus negative, not because they resist infection but because they had a lower cumulative exposure to *Mtb*. We therefore defined "resisters" as miners who were QFT-plus-negative and had a zero TST response.

## Statistical methods

The statistical analysis was conducted using STATA (version 15, StataCorp LP, College Station, Texas). Descriptive statistics were used to assess the proportion of gold miners that were TB uninfected and met the definition of resisters based on high cumulative exposure and their QFT-Plus and TST results as described above. Logistic regression was used to identify epidemiological factors associated with being TB uninfected. Factors in the univariable analysis with evidence of an association with the outcome (p value from the likelihood ratio test, 0.2) and, *a priori*, age group were included in the multivariable model. Given the relatively small number of outcomes (91), the adjusted model was restricted to at most nine parameters to avoid estimation problems due to data sparsity. To increase the likelihood of identifying epidemiologic factors, a further analysis was done, restricted to participants of Black/African ethnicity who, in this setting are more likely to have high cumulative exposure to *Mtb*. To identify epidemiological factors associated with being a resister, a sensitivity analysis using the stricter definition of being TB uninfected and restricted to participants of Black/African ethnicity was conducted.

## Results

### Participants

From 10 July 2015 to 12 December 2016, we offered the pre-screen to 25,637 miners of whom 17,030 agreed (Fig 1). At the pre-screen 13,496 (79.2%) did not meet the criteria. Of the 3,534

**Table 1. Baseline demographic characteristics.**

| | | Participants Enrolled | Participant enrolled with a QFT-plus result | Participants enrolled with a QFT-plus & TST result* |
|---|---|---|---|---|
| | N | 349 | 307 | 235 |
| **Age, years** | Median (IQR) | 48 (45–53) | 48 (44–53) | 48 (44–53) |
| **Age, years (grouped)** | <45 | 87 (24.9%) | 79 (25.7%) | 61 (26.0%) |
| | 45–49 | 114 (32.7%) | 97 (31.6%) | 72 (30.6%) |
| | ≥50 | 148 (42.4%) | 131 (42.7%) | 102 (43.4%) |
| **Sex** | Male | 344 (98.6%) | 303 (98.7%) | 231 (98.3%) |
| **BCG scar** | No | 84 (24.1%) | 76 (24.8%) | 58 (24.7%) |
| | Yes | 253 (72.5%) | 222 (72.3%) | 172 (73.2%) |
| | Indeterminate | 12 (3.4%) | 9 (2.9%) | 5 (2.1%) |
| **Country of birth** | South Africa | 233 (66.8%) | 206 (67.1%) | 158 (67.2%) |
| | Lesotho | 59 (16.9%) | 51 (16.6%) | 38 (16.2%) |
| | Mozambique | 37 (10.6%) | 32 (10.4%) | 26 (11.1%) |
| | Other | 20 (5.7%) | 18 (5.9%) | 13 (5.5%) |
| **Ethnicity** | Black/African | 321 (92.0%) | 281 (91.5%) | 218 (92.8%) |
| **Marital Status** | Married | 313 (89.7%) | 274 (89.3%) | 210 (89.4%) |
| **Hostel** | Non-mining accommodation | 170 (48.7%) | 156 (50.8%) | 123 (52.3%) |
| | Mining hostel¶ | 100 (28.7%) | 84 (27.4%) | 58 (24.7%) |
| | Other mining accommodation | 79 (22.6%) | 67 (21.8%) | 54 (23.0%) |
| **Years worked underground** | Median (IQR) | 24 (18–28) | 24 (18–28) | 24 (17–28) |
| **(grouped)** | <20 | 101 (28.9%) | 88 (28.7%) | 68 (28.9%) |
| | 20–29 | 184 (52.7%) | 164 (53.4%) | 122 (51.9%) |
| | ≥30 | 64 (18.3%) | 55 (17.9%) | 45 (19.2%) |
| **Sleeping arrangement** | Alone | 51 (14.6%) | 46 (15%) | 34 (14.5%) |
| | 1 person | 206 (59.0%) | 183 (59.6%) | 140 (59.6%) |
| | > 1 person | 92 (26.4%) | 78 (25.4%) | 61 (26.0%) |
| **Occupation** | Skilled/other | 78 (22.4%) | 70 (22.8%) | 46 (19.6%) |
| | Unskilled | 271 (77.7%) | 237 (77.2%) | 189 (80.4%) |

IQR interquartile range; QFT-Plus QuantiFERON-TB Gold-Plus; TST tuberculin skin test; BCG Bacille Calmette-Guérin.

* QFT-Plus and TST result within 7 days of each other.

¶ Mine hostel accommodation is provided by mining companies for the mineworkers they employ and often the miners live in crowded conditions.

(21%) of miners satisfied the pre-screen, 2,980 (84%) were offered consent for the full screen of whom 1,749 (59%) consented (Fig 1). Following the full screen, 1,400 (/1749; 80%) were screened out; 349 (20%) met the inclusion criteria and were included in the analysis (Fig 1). Overall, 307/349 (88%) participants had a QFT-Plus result and 235/349 (67%) had QFT-Plus and TST result within seven days of each other.

## Demographic characteristics of participants

Of the 349 participants included in the analysis, the median age was 48 years (interquartile range [IQR] 45–53 years), median time in the workface was 24 years (IQR 18–28 years) and 99% (344) were male. Overall, 92% (321) were of Black/African ethnicity, 72% (253) had a BCG scar, 67% (233) were born in South Africa and less than a third currently lived in a mine hostel (29%; 100; Table 1). Of Black/Africans 83% were in unskilled jobs vs 18% among non-

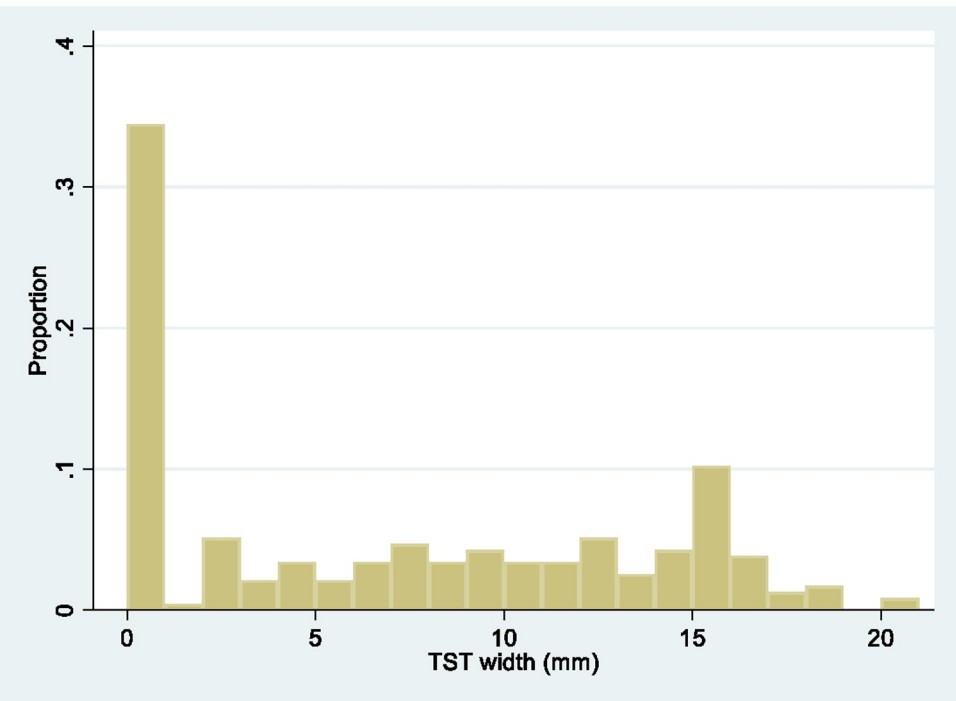

**Fig 2. Distribution of TST responses (n = 235).** TST = tuberculin skin test.

Black/Africans. The demographics of miners that had a QFT-Plus result (n = 307) or QFT-Plus and TST results (n = 235) was similar to the overall group (n = 349) (Table 1).

## QuantiFERON-Plus results and TST distribution

Thirty-percent (91/307) of participants with a baseline QFT-Plus result had a negative QFT-Plus result. Of the 307 participants, 281 (92%) were of Black/African ethnicity of which 25% (71/281) had a negative QFT-Plus result. Of the 235 participants that had both QFT-plus and TST results, 35 (15%) were QFT-Plus negative/zero response on TST, 101 (43%) were QFT-Plus positive/TST ≥5mm and 99 (42%) had discordant results. The TST distribution among 235 participants with both results, 34% (81), 11% (26) and 54% (128) had 0mm, >0mm to <5mm, and ≥5mm TST response, respectively (Fig 2). Among the 218 miners of Black/African ethnicity with QFT-Plus and TST results (Table 1), 23 (11%) were QFT-Plus negative/zero TST response and were defined as resisters.

Overall of 91 participants with a QFT plus negative result at baseline, 47 (51.6%) had a QFT plus result at the 12 month follow-up visit, of whom 4 (8.5%) had a positive QFT result. Further, of the 35 participants satisfying the strict definition of being TB uninfected (QFT-negative and TST 0mm), 57% (n = 20) had a QFT result at the 12 month follow-up visit, of whom all were QFT-negative. These results were not used further in this analysis.

## Epidemiological factors associated with being TB uninfected

Among those who had a baseline QFT-plus result (n = 307), miners who were older, of Black/African ethnicity, lived in a hostel, were unskilled workers and had a BMI ≥30, were less likely to be TB uninfected (Table 2). A reverse trend was observed with the prevalence of being TB uninfected higher with increasing number of people sleeping in the same room (Table 2A). In

**Table 2. Factors associated with being TB uninfected as defined by being QFT-Plus negative, among miners attending for annual medical examination: 2(a) overall (n = 307); and 2(b) restricted to participants of Black/African ethnicity (n = 281).**

| | Variable | QFT-Plus Negative (n/N) | Unadjusted OR (95% CI) | P Value | Adjusted OR* (95% CI) | P Value |
|---|---|---|---|---|---|---|
| **2(a) overall (n = 307)** | | | | | | |
| **Age, years** | <45 | 39.2% (31/79) | 1 | 0.10 | 1 | 0.21 |
| | 45–49 | 25.8% (25/97) | 0.54 (0.28–1.02) | | 0.58 (0.30–1.12) | |
| | ≥50 | 26.7% (35/131) | 0.56 (0.31–1.02) | | 0.63 (0.34–1.17) | |
| **Sex** | Male | 30.0% (91/303) | | 0.32** | | |
| | Female | 0.0% (0/4) | | | | |
| **BCG Scar** | No | 26.3% (20/76) | 1 | 0.46 | | |
| | Yes | 30.7% (71/231) | 1.2 (0.69–2.22) | | | |
| **Country** | South Africa | 30.6% (63/206) | 1 | 0.62 | | |
| | Lesotho | 23.5% (12/51) | 0.70 (0.34–1.42) | | | |
| | Mozambique | 28.1% (9/32) | 0.89 (0.39–2.03) | | | |
| | Other | 38.9% (7/18) | 1.44 (0.54–3.90) | | | |
| **Ethnicity** | Black/African | 25.3% (71/281) | 1 | <0.001 | | |
| | Other | 76.9% (20/26) | 9.9 (3.8–25.5) | | | |
| **Marital Status** | Married | 28.8% (79/274) | 1 | 0.38 | | |
| | Other | 36.4% (12/33) | 1.4 (0.66–3.00) | | | |
| **Live** | Non-mining accommodation[1] | 34.0% (53/156) | 1 | 0.04 | 1 | 0.15 |
| | Mining hostel[2] | 19.1% (16/84) | 0.46 (0.24–0.87) | | 0.53 (0.27–1.02) | |
| | Other mining accommodation | 32.8% (22/67) | 0.95 (0.52–1.75) | | 0.91 (0.48–1.70) | |
| **Years worked underground** | <20 years | 30.7% (27/88) | 1 | 0.75 | | |
| | 20–29 years | 30.5% (50/168) | 0.99 (0.56–1.74) | | | |
| | ≥30 years | 25.5% (14/55) | 0.77 (0.36–1.64) | | | |
| **Sleeping arrangement** | Alone | 17.4% (8/46) | 1 | 0.06 | | |
| | 1 person | 29.5% (54/183) | 1.99 (0.87–4.54) | | | |
| | > 1 person | 37.2% (29/78) | 2.81 (1.15–6.85) | | | |
| **Occupation** | Skilled/other | 44.3% (31/70) | 1 | <0.01 | 1 | 0.005 |
| | Unskilled | 25.3% (60/237) | 0.43 (0.24–0.74) | | 0.43 (0.24–0.77) | |
| **BMI grouped, kg/m²** | 18–24.9 | 37.3% (28/75) | 1 | 0.09 | 1 | 0.05 |
| | 25–29.9 | 30.8% (41/133) | 0.78 (0.41–1.36) | | 0.73 (0.39–1.35) | |
| | ≥30 | 22.2% (22/99) | 0.48 (0.25–0.93) | | 0.42 (0.21–0.85) | |
| **2(b) restricted to participants of Black/African ethnicity (n = 281)** | | | | | | |
| **Age, years** | <45 | 32.9% (23/70) | 1 | 0.25 | 1 | 0.41 |
| | 45–49 | 23.6% (21/89) | 0.63 (0.31–1.27) | | 0.66 (0.32–1.34) | |
| | ≥50 | 22.1% (27/122) | 0.58 (0.30–1.12) | | 0.66 (0.34–1.30) | |
| **Sex** | Male | 25.6% (71/277) | | 0.58* | | |
| | Female | 0.0% (0/4) | | | | |
| **BCG Scar** | No | 20.0% (14/70) | 1 | 0.23 | | |
| | Yes | 27.0% (57/211) | 1.48 (0.77–2.86) | | | |
| **Country** | South Africa | 24.7% (45/182) | 1 | 0.91 | | |
| | Lesotho | 23.5% (12/51) | 0.94 (0.45–1.94) | | | |
| | Mozambique | 28.1% (9/32) | 1.19 (0.51–2.76) | | | |
| | Other | 31.2% (5/16) | 1.38 (0.46–4.20) | | | |
| **Ethnicity** | Black/African | | NA | | | |
| | Other | | | | | |
| **Marital Status** | Married | 24.6% (62/252) | 1 | 0.46 | | |
| | Other | 31.0% (9/29) | 1.38 (0.60–3.19) | | | |

*(Continued)*

**Table 2.** (Continued)

| | Variable | QFT-Plus Negative (n/N) | Unadjusted OR (95% CI) | P Value | Adjusted OR* (95% CI) | P Value |
|---|---|---|---|---|---|---|
| **Live** | Non-mining accommodation[1] | 25.8% (34/132) | 1 | 0.18 | 1 | 0.26 |
| | Mining hostel[2] | 19.0% (16/84) | 0.68 (0.35–1.33) | | 0.71 (0.36–1.41) | |
| | Other mining accommodation | 32.3% (21/65) | 1.38 (0.72–2.63) | | 1.38 (0.70–2.70) | |
| **Years worked underground** | <20 years | 25.0% (20/80) | 1 | 0.54 | | |
| | 20–29 years | 27.3% (41/150) | 1.13 (0.61–2.10) | | | |
| | ≥30 years | 19.6% (10/51) | 0.73 (0.31–1.72) | | | |
| **Sleeping arrangement** | Alone | 15.6% (7/45) | 1 | 0.22 | | |
| | 1 person | 26.6% (45/169) | 1.97 (0.82–4.73) | | | |
| | > 1 person | 28.4% (19/67) | 2.15 (0.82–5.64) | | | |
| **Occupation** | Skilled/other | 32.6% (16/49) | 1 | 0.20 | 1 | 0.34 |
| | Unskilled | 23.7% (55/232) | 0.64 (0.33–1.25) | | 0.71 (0.35–1.42) | |
| **BMI grouped, kg/m²** | 18–24.9 | 35.6% (26/73) | 1 | 0.02 | 1 | 0.04 |
| | 25–29.9 | 24.6% (30/122) | 0.59 (0.31–1.11) | | 0.58 (0.31–1.12) | |
| | ≥30 | 17.4% (15/86) | 0.38 (0.18–0.80) | | 0.39 (0.18–0.82) | |

* Adjusted for age (a priori), living quarters (live), occupation and BMI.

** from Fishers exact test; [1] this comprises informal and formal dwellings which it is not possible to distinguish between; [2] Mine hostel accommodation is provided by mining companies for the mineworkers they employ and often the miners live in crowded conditions.

QFT-Plus QuantiFERON-TB Gold-Plus; OR Odds ratio; CI confidence interval.

an adjusted analysis, we omitted ethnicity and sleeping arrangement, due to collinearity with occupation and type of living quarters, respectively, and found that unskilled workers and miners with BMI ≥30 were less likely to be TB uninfected (Table 2A).

In an analysis restricted to participants of Black/African ethnicity (71/281; 25%) were QFT-Plus negative), miners that had BMI ≥30 were less likely to be to be TB uninfected (Odds ratio [OR] 0.38; 95% confidence interval [CI] 0.18–0.80; p = 0.02; Table 2B). In an adjusted analysis, we omitted ethnicity and sleeping arrangement, due to collinearity with occupation and type of living quarters, respectively, the association with BMI remained similar (Table 2B).

In a sensitivity analysis based on 136 miners that had concordant QFT-Plus and TST results (Fig 1), 26% (35/136) were QFT-Plus negative/zero TST response and were classified as TB uninfected (resister). Black/African miners and unskilled miners were less likely to be TB uninfected (Table 3). A multivariable analysis was not conducted, due to the small number of outcomes. When restricted to 123 miners of Black/African Ethnicity, 23 (19%) were QFT-Plus negative and had zero response on TST. Where miners live was associated with being a resister (p = 0.04). Miners living in a hostel were less likely to be resisters, though the CI were wide (Odds ratio [OR] 0.86; 95% confidence interval [CI] 0.0.24–3.03). Again a multivariable analysis was not conducted, due to the small number of outcomes.

## Discussion

Our cross-sectional study assessed epidemiological characteristics associated with being TB uninfected among gold miners in high TB burden settings. Despite prolonged periods of exposure to tuberculosis, we identified a small proportion of individuals that were TB uninfected. Although rare, we identified individuals meeting the definition of "resisters" that were QFT negative and had a zero TST response. We identified factors associated with being TB uninfected, including ethnicity, occupation (skilled versus unskilled labour) and BMI. The association between ethnicity and being TB uninfected most likely reflects higher exposure among

**Table 3. Factors associated with being TB uninfected defined as being QuantiFERON-TB Gold-Plus negative and tuberculin skin test zero mm response† among miners attending for annual medical examination (n = 136) and restricted to Black/African miners (n = 123).**

| | Variable | All miners (n = 136) | | | Black/African miners (n = 123) | | |
|---|---|---|---|---|---|---|---|
| | | QFT-Plus/TST Negative (n/N) | Unadjusted OR (95% CI) | P-Value | QFT-Plus/TST Negative (n/N) | Unadjusted OR (95% CI) | P-Value |
| Age, years | <45 | 38.9% (14/36) | 1 | 0.09 | 31.2% (10/33) | 1 | 0.11 |
| | 45–49 | 16.7% (6/36) | 0.31 (0.10–0.95) | | 11.8% (4/34) | 0.29 (0.08–1.06) | |
| | ≥50 | 23.4% (15/64) | 0.48 (0.20–1.17) | | 15.9% (9/57) | 0.41 (0.14–1.56) | |
| Sex | Male | 26.5% (35/132) | | 0.57* | 19.3% (23/119) | | >0.99* |
| | Female | 0.0% (0/4) | | | 0.0% (0/4) | | |
| BCG Scar | No | 29.4% (10/34) | 1 | 0.57 | 17.2% (5/29) | 1 | 0.82 |
| | Yes | 24.5% (25/102) | 0.78 (0.33–1.85) | | 19.1% (18/94) | 1.14 (0.38–3.39) | |
| Country | South Africa | 26.7% (24/90) | 1 | 0.54 | 16.7% (13/78) | 1 | 0.66 |
| | Lesotho | 16.7% (4/24) | 0.55 (0.17–1.77) | | 16.7% (4/24) | 1.00 (0.29–3.41) | |
| | Mozambique | 26.7% (4/15) | 1.00 (0.29–3.44) | | 26.7% (4/15) | 1.82 (0.50–6.61) | |
| | Other | 42.9% (3/7) | 2.06 (0.43–9.9) | | 33.3% (2/6) | 2.5 (0.41–15.1) | |
| Ethnicity | Black/African | 18.7% (23/123) | 1 | <0.001 | NA | | |
| | Other | 92.3% (12/13) | 52.2 (6.45–421) | | | | |
| Marital Status | Married | 23.6% (29/123) | 1 | 0.09 | 17.0% (19/112) | 1 | 0.15 |
| | Other | 46.1% (6/13) | 2.78 (0.86–8.93) | | 36.4% (4/11) | 2.80 (0.74–10.5) | |
| Live | Non-mining accommodation | 27.6% (21/76) | 1 | 0.09 | 14.3% (9/63) | 1 | 0.04 |
| | Mine hostel | 12.5% (4/32) | 0.37 (0.12–1.20) | | 12.5% (4/32) | 0.86 (0.24–3.03) | |
| | Other mining accommodation | 35.7% (10/28) | 1.46 (0.58–3.66) | | 35.7% (10/28) | 3.33 (1.17–9.49) | |
| Years worked underground | <20 years | 27.9% (12/43) | 1 | 0.31 | 20.5% (8/39) | 1 | 0.24 |
| | 20–29 years | 28.8% (19/66) | 1.04 (0.44–2.45) | | 22.0% (13/59) | 1.10 (0.40–2.95) | |
| | ≥30 years | 14.8% (4/27) | 0.45 (0.13–1.57) | | 8.00% (2/25) | 0.34 (0.07–1.74) | |
| Sleeping arrangement | Alone | 12.5% (3/24) | 1 | 0.07 | 12.5% (2/23) | 1 | 0.33 |
| | 1 person | 24.4% (19/78) | 2.25 (0.60–8.40) | | 24.4% (15/73) | 2.71 (0.57–12.9) | |
| | > 1 person | 38.2% (13/34) | 4.33 (1.08–17.5) | | 38.2% (6/27) | 3.00 (0.54–16.6) | |
| Occupation | Skilled/other | 43.3% (13/30) | 1 | 0.02 | 23.8% (5/21) | 1 | 0.52 |
| | Unskilled | 20.7% (22/106) | 0.34 (0.14–0.81) | | 17.7% (18/102) | 0.69 (0.22–2.11) | |
| BMI grouped, kg/m² | 18.5–24.9 | 29.6% (8/27) | 1 | 0.31 | 26.9% (7/26) | 1 | 0.31 |
| | 25–29.9 | 29.7% (19/64) | 1.00 (0.37–2.68) | | 19.6% (11/56) | 0.66 (0.22–1.97) | |
| | ≥30 | 17.8% (8/45) | 0.51 (0.17–1.58) | | 12.2% (5/41) | 0.38 (0.11–1.35) | |

* from Fishers exact test.

QFT-Plus QuantiFERON-TB Gold-Plus; CI confidence interval ; NA not applicable.

† compared to being TB infected defined as QFT-Plus positive and TST induration ≥5mm.

miners of Black/African origin due to the historic working and living conditions in the gold mining settings. Similarly, occupation level reflects the lower exposure among skilled versus unskilled miners, with unskilled miners more likely to work underground where exposure to dust and silica is high. Exposure to silica dust has long been associated with pulmonary TB [18–21]. Among this cohort, miners of Black/African origin were more likely to be unskilled workers (83% versus 18% among those non- Black/African). Normal weight (BMI 18.5 to <25) and overweight (BMI 25 to <30) were more likely to be TB uninfected than obese miners (BMI ≥30). Although undernutrition (BMI<18.5) is well associated with tuberculosis,

association with obesity is not known. An unexpected reverse trend was observed with sleeping arrangement and it is unclear if the question was understood by the participants. However, when restricted to groups with high cumulative exposure, the numbers were small and it is uncertain if the observed results are due to the small sample size.

This is one of two studies describing the existence of this phenotype in populations highly exposed to *Mtb* infection, other than household contacts. Our data, adds to the body of evidence supporting the existence of individuals who resist *Mtb* infection despite high cumulative exposure and will provide data to further characterise genetic and immunological factors potentially associated with resistance in this population.

We found a low proportion (11%) of resisters in South African Gold miners who were highly exposed to TB through proxies of age and workforce and using a strict definition of being negative on both tests of infection. This was consistent with the cumulative resister proportion range of 0%-35% reported by Simmons *et al* in a recent review of 10 historical studies [11]. More recently in a multi-country study conducted among household contacts of multidrug resistant TB cases in Botswana, Brazil, Haiti, India, Kenya, Peru, South Africa and Thailand, 15% of household contacts with high exposure to TB were resistant to Mtb infection, [12]. The proportion of resisters in our study was also comparable to that among household contacts in Uganda (12%) and higher than reported in India (7%) respectively [13, 22]. As in our study, the diagnosis of a resister in the multi-country study was based on both QFT-Plus and TST where a negative results was defined by QFT-Plus negative/TST 0mm. However the initial Uganda studies used TST only to classify *M. tuberculosis* infection status and less stringent cut-off of <10mm in HIV negatives and <5mm in HIV positives and in the India study a TST cut off of <5mm and IGRA were used [22]. After 2 years of follow-up in the Uganda study, 10.7% of household contacts were persistently negative on TST [14]. More recently a longer-term follow up has been completed in a subset of the original household contacts enrolled in the study and repeat test of LTBI done using Quantiferon-TB gold (QFT-TB Gold) and TST at a median of 9.5 years post known exposure to an index case [9]. Notably of 162 resisters, 82 remained negative on QFT-TB Gold and TST of which 91.7% had a TST of 0mm. As with other studies, we did not find any definitive epidemiological characteristics associated with the resister phenotype.

Our study has several limitations. Firstly, unlike in the household contact studies, our study did not have a defined source case, rather we used the period working in gold mining settings as a proxy for prolonged exposure to *Mtb*. The longer the cumulative exposure the more likely a negative result from the two tests of infection, will reflect the resister phenotype. Studies done in household contacts, of individuals with microbiologically confirmed TB had well characterised source case and a precise measure of intensity and duration of exposure [8].

Secondly, we included a heterogeneous group of individuals of different ethnic backgrounds who had worked in the gold mines for ≥15 years and we characterized them as being highly exposed. By doing so we inadvertently included participants from other ethnic groups that had a much lower cumulative TB exposure given their socio-demographic status. To improve our definition of high exposure to Mtb infection so that all miners had similar exposures, we restricted our analysis to Black/African miners. Data from early studies suggests that Black African miners have higher cumulative exposure due to a number of factors including dense working and living conditions and circular migration outside the mines [19, 23, 24]. Thirdly, the definition of a resister relies on an indirect approach of measuring adaptive immune response to *Mtb* infection. Non-conversion in TST and IGRA could reflect inadequate exposure, involvement of innate immunity or other adaptive immune mechanisms other than IFN-. Fourthly, the tuberculin test uses purified protein derivative antigens, which are not specific to *Mtb*. Though highly sensitive TST has decreasing specificity with false

positives and cross reactions to previous BCG vaccination and nontuberculous mycobacteria (NTM). Previous studies in this gold mining population have shown high prevalence of disease due to NTM [18]. To increase specificity we used both TST and QFT-Plus, which measures interferon gamma release from antigen specific CD4+ T cells following stimulation with Mtb antigens. Adding two tests on its own presents a challenge. Agreement between TST and IGRA has been shown to range from 60–80% and test discordance is common in individuals with Mtb infection [25, 26]. In this study discordant results were reported in 42% of participants who has both TST and QFT plus results and these were excluded from the analysis. Fifthly, due to low numbers of individuals available to provide a sample for Mtb testing, we did not include follow up data in this analysis and as such we might have incorrectly classified individuals recently infected with Mtb. As observed in household contacts in Uganda, more than half of exposed individuals were TST negative at enrolment and converted their TST within 3–6 months [14, 22]. Unfortunately our target sample size was not reached resulting in limited adjusted analyses and reduced power to detect risk factors for TB infection. Lastly, the inability to adjust for confounding in the sensitivity analysis is a limitation and the unadjusted results might not reflect the actual relationship.

Potential cellular mechanisms underlying the clinical phenomenon of resistance to Mtb infection despite high exposure, were identified in household contacts in Uganda. Data from a genome scan study using genotypes from household contacts in Uganda, including individuals who have persistently negative TSTs despite prolonged exposure to infectious TB cases identified linkage between this phenotype and novel chromosomal loci [10]. In this study genetic factors were found to play a key role in *Mtb* infection and the interactions differed according to the outcome of the infection [10]. Furthermore immunological studies comparing cellular responses in *Mtb* susceptible and resistant household contacts in Uganda identified the role of histone deacetylase function in the innate immune response to Mtb infection [27]. More recently distinct gene expression profiles in monocytes from two distinct highly exposed cohorts from Uganda and South Africa were identified in individuals who resist Mtb infection [28]. The South African data reported was from this study.

## Conclusions

A substantial proportion of Black/African miners remain uninfected despite prolonged cumulative exposure to *Mtb*. In keeping with other studies we did not identify definitive epidemiological factors associated with being a resister. Ongoing studies looking at biological processes that may be fundamentally different between individuals who resist infection and those with latent MTB infection remains important.

## Acknowledgments

We thank the thousands of participants who consented to taking part in the study. We are thankful to the many stakeholders for their support for the study to be implemented: The North West Department of Health, National Union of Mine Workers and Anglo Gold Ashanti for their support for the study to be implemented. We appreciate the commitment and efforts of the HETU study teams in enrolling and interviewing participants, at the occupational health centre in North West Province, the laboratory and data management team for supporting the study.

## Author Contributions

**Conceptualization:** Violet N. Chihota, Pholo Maenetje, Thomas R. Hawn, Robert Wallis, Alison D. Grant, Gavin J. Churchyard, Katherine Fielding.

**Data curation:** Katherine Fielding.

**Formal analysis:** Raoul Mansukhani, Katherine Fielding.

**Funding acquisition:** Gavin J. Churchyard.

**Methodology:** Violet N. Chihota, Pholo Maenetje, Thomas R. Hawn, Robert Wallis, Alison D. Grant, Gavin J. Churchyard, Katherine Fielding.

**Project administration:** Violet N. Chihota, Thobani Ntshiqa, Kavindhran Velen.

**Supervision:** Violet N. Chihota, Kavindhran Velen, Alison D. Grant, Gavin J. Churchyard, Katherine Fielding.

**Visualization:** Raoul Mansukhani, Katherine Fielding.

**Writing – original draft:** Violet N. Chihota.

**Writing – review & editing:** Violet N. Chihota, Thobani Ntshiqa, Pholo Maenetje, Raoul Mansukhani, Kavindhran Velen, Thomas R. Hawn, Robert Wallis, Alison D. Grant, Gavin J. Churchyard, Katherine Fielding.

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
