## [Decision Letter · Decision Letter 0]

27 Apr 2021

PONE-D-21-05516

Epidemiological characteristics associated with resistance to Mycobacterium tuberculosis infection in highly TB exposed South African gold miners

PLOS ONE

Dear Dr. Chihota,

Thank you for submitting your manuscript to PLOS ONE. After careful consideration, we feel that it has merit but does not fully meet PLOS ONE’s publication criteria as it currently stands. Therefore, we invite you to submit a revised version of the manuscript that addresses the points raised during the review process.

Please submit your revised manuscript. If you will need significantly more time to complete your revisions, please reply to this message or contact the journal office at plosone@plos.org. Please include the following items when submitting your revised manuscript:

We look forward to receiving your revised manuscript.

Kind regards,

Frederick Quinn

Academic Editor

PLOS ONE

Journal Requirements:

Reviewers' comments:

Reviewer's Responses to Questions

**Comments to the Author**

1. Is the manuscript technically sound, and do the data support the conclusions?

Reviewer #1: No

Reviewer #2: Yes

Reviewer #3: Yes

2. Has the statistical analysis been performed appropriately and rigorously? 

Reviewer #1: Yes

Reviewer #2: Yes

Reviewer #3: Yes

3. Have the authors made all data underlying the findings in their manuscript fully available?

Reviewer #1: Yes

Reviewer #2: Yes

Reviewer #3: No

4. Is the manuscript presented in an intelligible fashion and written in standard English?

Reviewer #1: Yes

Reviewer #2: Yes

Reviewer #3: Yes

5. Review Comments to the Author

Reviewer #1: NE-D-21-05516

Full Title: Epidemiological characteristics associated with resistance to Mycobacterium tuberculosis infection in highly TB exposed South African gold miners

Major

South African gold miners are heavily exposed to Mycobacterium tuberculosis. It is therefore a suitable population to identify individuals with a “resister phenotype”.

I read the study as having three components, the first two to some extent obscured by the emphasis on the third.

1. The first is the descriptive question of the prevalence of negative LTBI status in gold miners, using two different tests, QFT-Plus and TST. This is interesting question in its own right and yields informative results.

Prevalence of QFT-plus negative varied by test and sample – 30% for the total sample, 25% for Black/African miners. For the subset who had results of both tests available, 34% had a TST of 0 mm induration and 45% a test < 5 mm (including zero). When a negative QFT-plus was combined with a negative TST test (“and”, not “and/or”) the prevalence was 15%. The most restrictive sample were Black/African miners with both tests negative (TST 0 mm) – yielding a prevalence of 11%.

This group were defined as “resisters” on the assumption that combining negative tests of the two studies yields the most specific subset. Individuals with discordant results were excluded. The authors reduction of the sample to the most likely subset of biological resisters makes sense, given component 2 below. However, the discordance between the two tests points to the complexity of the phenomenon of LTBI as a biological and epidemiological concept. As one example, IGRAs are characterised by regression from positive to negative. The authors might want to comment on these issues.

Regarding the Discussion, the summary of other studies of LTBI negative groups is informative, showing that the prevalence among gold miners is similar to or within the range of findings from elsewhere.

2. The second component identifies a subset of this population for further laboratory studies of immunological or other markers which might illuminate to the biological basis of resistance to TB infection.

3. The third component, which is presented in the title and text as the main substance of the article is an attempt to model the predictors of this reduced subset of zero reactors. This strikes me as a more problematic analysis.

It would be useful for the authors to provide some rationale for this component and some a priori consideration (possibly with hypotheses) of the predictor factors available for study.

Figure 1 indicates that the final sample of 235 was obtained from a sampling frame of 34 049 miners. I believe it is important for authors to explain the selection strategy in much more detail. The largest proportion excluded were those who for one reason or another were not available for screening or follow up or did not agree to participate or consent. A second group were subject to exclusion criteria indicative of past or current TB, IPT or HIV. However, the reasons for the remaining criteria: age < 33 years, employment < 15 years, chronic medication, and BMI < 18.5 need some explanation.

The one implication of this extensive reduction is loss of power, and generally wide confidence intervals, making some of the suggestive results statistically indeterminate. The other implication is that, given the interrelationship of most of the factors in the model, it is very difficult to know what selection biases have been introduced.

Interpretation is also limited by lack of rationale for the modelling strategy. For example, if only miners with > 15 years of employment were included to ensure that the the final sample had a high degree of infection opportunity, what purpose is served by stratifying this group further?

The authors should consider re-framing the write-up somewhat to rebalance the components referred to above. This would also entail a title change as I think the current title offers more than the article can deliver.

The limitations section needs to be expanded, as I think these go well beyond the items included in the current version.

Minor

p. 7, lines 129-130; p. 14, lines 280-281: Did any of the resister group convert during this 12 month follow-up? This would have tested the stability of resister status. (Were there any results from this follow-up?)

p. 4 lines 67-71: The sentence starting “Differences in..” is a non-sequitur. I suggest moving it elsewhere and starting the last sentence (“In studies of household..”) with “However..”.

p. 5, line 72: What is the comparator for the risk of Mtb among gold miners in South Africa? Is there a non-mining reference population that can be used?

Tables 2 and 3: We are so used to thinking of infection as the effect that interpreting the odds ratio as a measure of “being TB uninfected..” is difficult, as in “less likely to be TB uninfected”. The authors should consider inverting the odds ratios. It would make the tables easier to interpret intuitively.

***

Reviewer #2: Manuscript by Chihota et al. explores epidemiological characteristics of South African goldminers who, despite high exposure to M. tuberculosis, remain uninfected. Understanding natural resistance to TB infection and TB disease is of great interest and importance to the field. They found a small group of individuals who were classified as resisters consistent with other studies conducted in high burden settings. In addition, the authors found some correlates to TB uninfected status, including ethnicity, occupation and whether the miners lived in or outside a mine hostel and BMI. The manuscript overall is well written. Below some comments for consideration:

- It would be helpful if the authors articulated the basis of an epidemiological exploration? While I can see the value in finding epidemiological signals tracking with resister status, it would be helpful to unpack the motivation for such a study.

- Line 137: “In this risk factor analysis…” Unclear what RF analysis they are referring to.

- Line 155: The authors indicated that Black/African miners have higher cumulative exposure histories than others. As such they performed a subgroup analysis. Wouldn’t other miners of non-Black/African ethnicity with similar duration of employment have a very high degree of M. tuberculosis exposure?

- The authors found ethnicity, occupation, living arrangements and BMI associated with uninfected status. Yet there was no attempt to explain how these factors are biologically or epidemiologically plausible. That is, what ultimately can we infer from this study other than the prevalence of resisters in this high burden setting?

- Line 264: This study as with others did not find epidemiological signals associated with resister status. Was this expected in this study?

Reviewer #3: Overall, the paper provides evidence for the “resistor” phenotype in tuberculosis. It supports limited previous research that suggests some people may be more resistant to tuberculosis infection. It also identifies a possible cohort for future studies on genetic and immunological factors associated with tuberculosis resistance. Although it did not find definitive epidemiological factors associated with resistance to tuberculosis, it provides important new evidence that some people may be resistance to infection through analysis of a group of highly exposed underground miners in South Africa. I think the research is important and needs to be published as there are profound implications of identifying resistor phenotypes if this in turn aids future identification of genetic and immunological factors that could lead to improved vaccines or therapeutics for tuberculosis. Overall, the methodology and statistical approach is sound, and the findings support the conclusions.

There were a few major issues for the authors to consider:

1. It was not clear how the results of TB testing were collated and analysed in the study. It seems three sets of TB tests were taken over the course of 12 months. I could not work out which timepoint for testing was used for determining “resistors” or was it the collation of all TB tests over 12 months that determined “resistors”? I presume any positive test over the 12 months would exclude someone from the stricter definition of a “resistor”? It would help to make this more explicit. Furthermore, were converters or reverters included in the “resistor” group over 12 months? For example: line 193 – was this 91 of 307 at baseline only – what about QFT-Plus results at 4 months and 12 months?

2. Its not clear why the study was set up as a longitudinal study and not cross-sectional, as there is no time to event data or hazard analysis in the methods?

3. The prevalence of “resistors” is for this group of highly selected miners and may not represent the broader population from which the participants were drawn. I do not think this was adequately addressed in the Discussion. There is no discussion of the healthy worker selection effect and whether selecting out unhealthy workers could provide a biased estimate of the prevalence of “resistors”?

Minor issues for the authors to consider:

1. Line 53: what about progression in HIV + miners?

2. Line 55: Its not clear what the rates of TB in the UG mining workforce in SA are compared to the rest of the world. Maybe make it more explicit for the reader that it’s the highest rates of TB in the world as this provides a clearer picture of the force of exposure in this group.

3. Line 79: Could it be that the other arm of the immune system is responding? Humoral rather than cellular immunity?

4. Line 101: Is the annual medical mandatory? Is there potential for selection bias?

5. Line 109: how was silicosis determined?

6. Line 134: The sample size calculated with 769 yet the final sample was only 349. There is no discussion on this as a possible study limitation in the Discussion.

7. Line 145/6: Is this for all TB results at 0,4 and 12 months?

8. Line 157: ditto

9. Line 170: why was no adjustment done on the restricted analysis?

10. Line 245: see the Major Comment number 3.

11. Table 2: Should the restricted group also have been adjusted?

12. Table 2: Would state the factors adjusted for in the footnote of the table.

13. Line 344: a few grammar errors in the references. If using abbreviations for journals rather use this throughout.

14. There is no discussion on the boosting effect and if this could have been a source of bias or confounding.

15. There is no discission on whether silicosis could have been acting as a confounder. There was mention that miners were excluded from the study if found to have silicosis but how was the diagnosis of silicosis determined? Could resistors have had much lower exposure to silica dust by virtue of their occupation?

16. There are no references to Figure 1 and 2 in the text.

There are no confidential comments to the editor.

6. PLOS authors have the option to publish the peer review history of their article (what does this mean?). If published, this will include your full peer review and any attached files.

Reviewer #1: No

Reviewer #2: No

Reviewer #3: **Yes: **Dr D Knight

---

## [Author Response · Author response to Decision Letter 0]

7 Oct 2021

Responses to Editor and reviewer comments

Comment 1: Please ensure that your manuscript meets PLOS ONE's style requirements, including those for file naming. The PLOS ONE style templates can be found at

Response: We can confirm that the manuscript meets the PLOS ONE formatting requirements

Comment 2: We note that you have stated that you will provide repository information for your data at acceptance. Should your manuscript be accepted for publication, we will hold it until you provide the relevant accession numbers or DOIs necessary to access your data. If you wish to make changes to your Data Availability statement, please describe these changes in your cover letter and we will update your Data Availability statement to reflect the information you provide.

Response: The data will be provided on the London School of Hygiene and Tropical Medicine Datacompass repository. The data are currently being processed and the following DOI has been reserved: https://doi.org/10.17037/DATA.00002424

Reviewer #1: NE-D-21-05516

Full Title: Epidemiological characteristics associated with resistance to Mycobacterium tuberculosis infection in highly TB exposed South African gold miners

Major Comments 

South African gold miners are heavily exposed to Mycobacterium tuberculosis. It is therefore a suitable population to identify individuals with a “resister phenotype”.

I read the study as having three components, the first two to some extent obscured by the emphasis on the third.

Reviewer 1 Comment 1: Prevalence of QFT-plus negative varied by test and sample – 30% for the total sample, 25% for Black/African miners. For the subset who had results of both tests available, 34% had a TST of 0 mm induration and 45% a test < 5 mm (including zero). When a negative QFT-plus was combined with a negative TST test (“and”, not “and/or”) the prevalence was 15%. The most restrictive sample were Black/African miners with both tests negative (TST 0 mm) – yielding a prevalence of 11%.

This group were defined as “resisters” on the assumption that combining negative tests of the two studies yields the most specific subset. Individuals with discordant results were excluded. The authors reduction of the sample to the most likely subset of biological resisters makes sense, given component 2 below. However, the discordance between the two tests points to the complexity of the phenomenon of LTBI as a biological and epidemiological concept. As one example, IGRAs are characterised by regression from positive to negative. The authors might want to comment on these issues. 

Regarding the Discussion, the summary of other studies of LTBI negative groups is informative, showing that the prevalence among gold miners is similar to or within the range of findings from elsewhere.

Response: We agree that there are complexities with testing for Mtb infection with the current tests that are available. Under the first version of the protocol, QFT-plus testing was done at enrolment and a second sample collected and TST placed 90 days post enrolment. This was subsequently simplified to have QFT-plus and TST only done at enrolment. To maximize our sample size for the analysis of being TB infected based on QFT alone we used the QFT test done at enrolment for participants enrolled under either version of the protocol. For the stricter definition of being TB infected, we used the TST and QFT-plus done within 7 days of each other, either at 90 days post enrolment or at enrolment for participants enrolled under the 1st or 2nd version of the protocol, respectively. We have made changes to the methods (line 175-177) to clarify this. 

We have also included limitations related to LTBI testing in the discussion section line 333-346 and addressed the following aspects:

• Indirect nature of LTBI tests 

• Specificity of TST 

• Challenges related to using two tests of infection 

Reviewer 1 comment 2: The second component identifies a subset of this population for further laboratory studies of immunological or other markers, which might illuminate to the biological basis of resistance to TB infection.

Response: Enrolled participants provided samples for further biological tests and these results are presented in a separate manuscript (Simmons JD et al.) which combines data from our study and a study conducted in Uganda; the paper identifies distinct gene expression profiles in monocytes. 

Reference: Simmons JD et al. Monocyte metabolic transcriptional programs associate with resistance to tuberculin skin test/interferon-γ release assay conversion. J Clin Invest. 2021 131(14):e140073. doi: 10.1172/JCI140073. 

We have referred to this manuscript in the discussion section line 362-365.

Reviewer 1 Comment 3: The third component, which is presented in the title and text as the main substance of the article is an attempt to model the predictors of this reduced subset of zero reactors. This strikes me as a more problematic analysis.

It would be useful for the authors to provide some rationale for this component and some a priori consideration (possibly with hypotheses) of the predictor factors available for study.

Response: A previous study by Hanifa et al to determine the prevalence of latent TB infection and risk factors for positive tuberculin skin test among gold miners (Int J Tuberc Lung Dis. 2009;13: 39-46), found high prevalence of LTBI. Despite the high LTBI prevalence in this population, a small proportion of miners had a negative TST. This study however was not designed to identify individuals that were highly exposed to Mtb and did not specifically characterise the resister phenotype with respective to TST and IGRA. 

In the current study, we did not have a priori considerations of the predictor factors; however, we hypothesised that it would be possible to identify TB uninfected individuals in a population highly exposed to Mtb. We have now added the following sentence in the introduction line 91-94.

“We hypothesized that some individuals that have been exposed to Mtb do not get infected and that, it will be possible to identify highly TB exposed, uninfected miners and any epidemiological factors associated with that phenotype versus being TB infected (TST>=5mm & QFT positive)”.

Reviewer 1 Comment 4: Selection strategy: Figure 1 indicates that the final sample of 235 was obtained from a sampling frame of 34 049 miners. I believe it is important for authors to explain the selection strategy in much more detail. The largest proportion excluded were those who for one reason or another were not available for screening or follow up or did not agree to participate or consent. A second group were subject to exclusion criteria indicative of past or current TB, IPT or HIV. However, the reasons for the remaining criteria: age < 33 years, employment < 15 years, chronic medication, and BMI < 18.5 need some explanation.

The one implication of this extensive reduction is loss of power, and generally wide confidence intervals, making some of the suggestive results statistically indeterminate. The other implication is that, given the interrelationship of most of the factors in the model, it is very difficult to know what selection biases have been introduced.

Interpretation is also limited by lack of rationale for the modelling strategy. For example, if only miners with > 15 years of employment were included to ensure that the final sample had a high degree of infection opportunity, what purpose is served by stratifying this group further?

The authors should consider re-framing the write-up somewhat to rebalance the components referred to above. This would also entail a title change as I think the current title offers more than the article can deliver.

The limitations section needs to be expanded, as I think these go well beyond the items included in the current version.

Response: We agree that the selection strategy needs to be clearly explained. The purpose of the HETU study was to identify individuals who had high cumulative exposure to TB (based on what we know about TB in this setting) yet resist TB infection, with the main goal of determining any epidemiological or biological factors (gene expression and immunological profiles) associated with this phenotype. In the current paper, we describe identification of individuals who resist infection despite high exposure to Mtb and explore epidemiological factors associated with this phenotype. Biological factors associated with the phenotype are described elsewhere (Simmons JD et al. Monocyte metabolic transcriptional programs associate with resistance to tuberculin skin test/interferon-γ release assay conversion. J Clin Invest. 2021 131(14):e140073. doi: 10.1172/JCI140073).

We recognised that we would need to screen a large number of individuals to identify these resisters. The final sample of 235 was obtained from a potential sampling frame of 34 049, representing the total daily attendances at Occupational Health from July 2015-December 2016. This number will include multiple visits by miners (as miners attend annual for medical exams) and gives a head count during the study enrolment period. 

Study screening was done in two phases: 

i. The pre-screen which excluded individuals who did not have prolonged exposure to TB, based on proxies of age and years in the workforce (age <33 yrs, worked on mine <15yrs), relevant in this setting; followed by (among those not excluded) the full-screen.

ii. The full-screen, done in five stages (see fig 1) whereby if an individual did not satisfy an earlier stage they were excluded immediately. The main purpose of this full-screen was to exclude those who either had a past history or current TB, or those at higher risk of TB disease (using markers such as IPT, on chronic medication, silicosis, HIV-positive, BMI <18.5). We also excluded individuals who refused to have blood drawn for further tests. 

Of those satisfying both screens (high cumulative exposure to TB based on proxies of age and years in the workforce, no TB disease or not being at higher risk of TB disease and agreed to have blood draw (n=349)) we then assessed for TB infection (using IGRA, TST) to identify individuals who resist TB infection. 

Of the total head count, we were able to offer the pre-screen to 25,637, of whom 17,030 agreed. At the pre-screen 13,496 (79.2%) did not meet these criteria. Of the remaining 3534, 1748 were offered to participate in the study and agreed to take part in the full-screen. Overall 349 (20%) satisfied the full-screen, of whom we have QFT-plus for 307 and a subset of 235 who also have TST.

We realise this was not clear in the manuscript and have made edits to the methods (line 114-144) to explain the screening process and have also edited Figure 1 to show the two screening phases clearly. 

Minor comments 

Reviewer 1 Comment 6: p. 7, lines 129-130; p. 14, lines 280-281: Did any of the resister group convert during this 12-month follow-up? This would have tested the stability of resister status. (Were there any results from this follow-up?)

Response: The study was designed to focus the 12-month follow up on individuals who had a negative QFT test at baseline. The 12-month follow up was not the purpose of the analysis presented. However we now report the results (line 237-241).

Overall of 91 participants with a QFT plus negative result at baseline, 47 (51.6%) had a QFT plus data at follow-up. Of the 47 who were negative at baseline, 4 (8.5%) had a positive QFT result at follow up. Further, of the 35 participants satisfying the strict definition of being TB uninfected (QFT-negative and TST 0mm), 57% (n=20) had a QFT result at the 12 month visit, of whom all were QFT-negative. 

We unfortunately had low completion of the follow up visit at 12 months with only 52% of the QFT negatives followed up.

Reviewer 1 Comment 7: p. 4 lines 67-71: The sentence starting “Differences in..” is a non-sequitur. I suggest moving it elsewhere and starting the last sentence (“In studies of household..”) with “However..”.

Response: We agree with the comment and have deleted the sentence on page 4

Reviewer 1 Comment 8: p. 5, line 72: What is the comparator for the risk of Mtb among gold miners in South Africa? Is there a non-mining reference population that can be used?

Response: In comparison to the general population in South Africa, gold miners have a greater risk of TB. We have made this explicit in line 76-77.

Reviewer 1 Comment 9: Tables 2 and 3: We are so used to thinking of infection as the effect that interpreting the odds ratio as a measure of “being TB uninfected.” is difficult, as in “less likely to be TB uninfected”. The authors should consider inverting the odds ratios. It would make the tables easier to interpret intuitively.

Response: Our primary outcome was being TB uninfected and with respect, we prefer to keep the analysis as displayed. We have not made any changes to the tables. The purpose of the study is to look at people who resist infection and changing the outcome to TB infected will be more confusing.

Reviewer #2: 

Manuscript by Chihota et al. explores epidemiological characteristics of South African goldminers who, despite high exposure to M. tuberculosis, remain uninfected. Understanding natural resistance to TB infection and TB disease is of great interest and importance to the field. They found a small group of individuals who were classified as resisters consistent with other studies conducted in high burden settings. In addition, the authors found some correlates to TB uninfected status, including ethnicity, occupation and whether the miners lived in or outside a mine hostel and BMI. The manuscript overall is well written. Below some comments for consideration:

Reviewer 2 Comment 1: It would be helpful if the authors articulated the basis of an epidemiological exploration? While I can see the value in finding epidemiological signals tracking with resister status, it would be helpful to unpack the motivation for such a study.

Response: As mentioned above, a previous study by Hanifa et al to determine the prevalence of latent TB infection and risk factors for positive tuberculin skin test among gold miners (Int J Tuberc Lung Dis. 2009;13: 39-46), found high prevalence of LTBI. The risk of infection with Mtb was associated with some epidemiological factors such as ethnicity, frequency of working underground, duration of employment, congregate working, living and social conditions. Despite the high LTBI prevalence in this population, a small proportion of miners had a negative TST. 

In the current study, we hypothesised that some individuals that have been exposed to Mtb are not infected (resist infection). Individuals that resist infection in the presence of high Mtb exposure have not been well characterised. TB infection is clinically silent and it is not certain whether an “exposed, TB uninfected” phenotype exists due to epidemiological or biological factors. 

We have now revised the introduction line 91-94. 

Reviewer 2, Comment 2: Line 137: “In this risk factor analysis…” Unclear what RF analysis they are referring to.

Response: In this study we were identifying factors associated with being TB uninfected and this section was referring to the analysis done to identify these. We have deleted the section headed “risk factor analysis” because we define the study outcome and our approach to identifying epidemiological factors associated with being uninfected in the section headed “study outcome and definitions”. 

Reviewer 2, Comment 3: Line 155: The authors indicated that Black/African miners have higher cumulative exposure histories than others. As such they performed a subgroup analysis. Wouldn’t other miners of non-Black/African ethnicity with similar duration of employment have a very high degree of M. tuberculosis exposure?

Response: Data from early studies in gold mining settings in South Africa suggests that miners of Black/African ethnicity have higher cumulative exposure compared to miners of non-Black/African ethnic groups due to congregate living and working conditions. We included this in the discussion line 326-331. Our data also suggests that miners of Black/African ethnicity are more likely to be employed as unskilled miners in comparison to other ethnic groups. We have now added this information in the results section, line 224-225.

Reviewer 2, Comment 4: The authors found ethnicity, occupation, living arrangements and BMI associated with uninfected status. Yet there was no attempt to explain how these factors are biologically or epidemiologically plausible. That is, what ultimately can we infer from this study other than the prevalence of resisters in this high burden setting?

Response: The HETU study was designed to identify individuals who resist Mtb infection among individuals identified as being highly exposed (high cumulative exposure to TB based on proxies of age and years in the workforce, no TB disease or being at lower risk of TB disease). Once identified, the intention was to explore whether there were any epidemiological factors associated with this phenotype (the data reported in this paper) and in separate analysis explore biological factors associated with this phenotype (data reported elsewhere - Simmons JD et al. Monocyte metabolic transcriptional programs associate with resistance to tuberculin skin test/interferon-γ release assay conversion. J Clin Invest. 2021 131(14):e140073. doi: 10.1172/JCI140073).

We defined being TB uninfected as having a negative QFT test result. We also used a stricter definition defined as being negative on both QFT and TST (0mm response) which we feel more confident aligns with being a resister. Being TB uninfected was associated with a few factors which are markers of exposure to TB infection. In the discussion section, line 271-291, we have now included a brief explanation of the plausibility of these associations. 

Reviewer 2, Comment 5: Line 264: This study as with others did not find epidemiological signals associated with resister status. Was this expected in this study?

Response: In our study we assess epidemiological characteristics of TB uninfected (negative on QFT plus) and resisters (QFT Plus negative and TST=0mm) among gold miners in high TB burden settings. We found that TB uninfected miners were rare and those meeting the definition of a resistors were more rare. As with other studies assessing this phenotype in household contacts, our study did not identify any epidemiological factors that were clearly associated with the resister phenotype. When we restricted our analysis to Black/African miners and using the definition of TST=0mm/QFT negative, the sample size was small (n=123 with 23 being classified as TB uninfected) and we could not say with certainty if the observed results were due to a small sample size. However, some of the effect estimates were in the same direction as the larger group, albeit wider confidence intervals. 

Reviewer #3: 

Overall, the paper provides evidence for the “resistor” phenotype in tuberculosis. It supports limited previous research that suggests some people may be more resistant to tuberculosis infection. It also identifies a possible cohort for future studies on genetic and immunological factors associated with tuberculosis resistance. Although it did not find definitive epidemiological factors associated with resistance to tuberculosis, it provides important new evidence that some people may be resistance to infection through analysis of a group of highly exposed underground miners in South Africa. I think the research is important and needs to be published as there are profound implications of identifying resistor phenotypes if this in turn aids future identification of genetic and immunological factors that could lead to improved vaccines or therapeutics for tuberculosis. Overall, the methodology and statistical approach is sound, and the findings support the conclusions.

There were a few major issues for the authors to consider:

Major comments 

Reviewer 3, Comment 1:

It was not clear how the results of TB testing were collated and analysed in the study. It seems three sets of TB tests were taken over the course of 12 months. I could not work out which timepoint for testing was used for determining “resistors” or was it the collation of all TB tests over 12 months that determined “resistors”? I presume any positive test over the 12 months would exclude someone from the stricter definition of a “resistor”? It would help to make this more explicit. Furthermore, were converters or reverters included in the “resistor” group over 12 months? For example: line 193 – was this 91 of 307 at baseline only – what about QFT-Plus results at 4 months and 12 months?

Response: As explained in the Methods section, under the first version of the protocol, we collected a blood sample for QFT at enrolment followed by another blood sample for QFT and TST placed 90 days after enrolment. This was simplified later to both QFT and TST done at enrolment. 

For the QFT-only outcome, to maximise sample size, we used the enrolment QFT measure for participants enrolled under either version of the protocol.

To define the resister phenotype we used results of test of infection (QFT and TST) at 90-day follow-up for those enrolled using the first version of the protocol and results of tests done as enrolment for those enrolled under the later version of the protocol. We have now made this clear in the Methods section headed “Study outcome and definitions” line 175-185.

The main analyses presented in the paper are based on these outcomes measured cross-sectionally. 

In addition, participants with a negative QFT result at enrolment, were followed-up 12 months post enrolment for a further QFT and TST. Of the 91 participants QFT-negative at enrolment, 52% (n=47) had a QFT result at the 12 month visit, of whom 4 were QFT-positive. Further, of the 35 participants satisfying the strict definition of being TB uninfected (QFT-negative and TST 0mm), 57% (n=20) had a QFT result at the 12 month visit, of whom all were QFT-negative. We have now revised the sentence in Methods line 165-167 to make clear who was followed 

up 12 months post enrolment and included data on this 12 month follow-up visit in the results (line 237-241). 

Reviewer 3, Comment 2

It’s not clear why the study was set up as a longitudinal study and not cross-sectional, as there is no time to event data or hazard analysis in the methods?

Response: The main HETU study was a longitudinal study. However even though participants were followed up at 90 days (for those enrolled using protocol version I) and 12 months, the main analyses presented used data collected at one time point (cross-sectionally) to classify the resisters.

We have now also included limited data on the 12-month follow-up (also see response to reviewer 3, comment 1), due to low ascertainment at the 12 month visit.

Reviewer 3, Comment 3

The prevalence of “resistors” is for this group of highly selected miners and may not represent the broader population from which the participants were drawn. I do not think this was adequately addressed in the Discussion. There is no discussion of the healthy worker selection effect and whether selecting out unhealthy workers could provide a biased estimate of the prevalence of “resistors”?

Response: In the HETU study we defined resisters from miners defined as highly exposed to TB and met the strict definition of being TB uninfected (QFT negative/TST=0mm). In household contact studies high exposure would mean living in household with an adult with TB. In our study we do not have a “point source” but have used age/years in workforce as proxies for sustained cumulative exposure – and also excluded those with history of TB/risk factors for TB. So in this highly selected group we then identify resisters, that is in addition they are negative for QFT and TST=0mm.

We never intended to measure the prevalence of resisters in this population. Interpreting 91/307 (30%) or 35/136 (26%) as such is complex based on the sampling frame. We rather were aiming to identify individuals that resist infection among individuals meeting the criteria of being highly exposed based on the proxies of age and years in the workforce and no evidence of TB disease or not being at higher risk of TB disease. We still believe, however, that our risk factor analysis for being TB uninfected (either definition) is of interest. 

Minor issues for the authors to consider:

Reviewer 3, Comment 4: Line 53: what about progression in HIV + miners?

Response: Immunocompromised persons with LTBI are at increased risk for progression to active TB and we have now made this clear in the introduction line 54-56. In our study, all individuals were HIV negative. 

Reviewer 3, Comment 5: Line 55: It is not clear what the rates of TB in the UG mining workforce in SA are compared to the rest of the world. Maybe make it more explicit for the reader that it’s the highest rates of TB in the world as this provides a clearer picture of the force of exposure in this group.

Response: Thank you we have now included some background on the burden of TB in the mines (line 59-62). 

Reviewer 3, Comment 6: Line 79: Could it be that the other arm of the immune system is responding? Humoral rather than cellular immunity?

Response: Thank you for the comment. It is possible that mechanisms that underly resistance to Mtb infection could reflect early clearance of the bacilli via innate immune responses that are activated prior to T cell responses or possibly other adaptive immune responses independent of IFN-ɣ that is measured but the tests of Mtb infection. We have included this as a limitation of the diagnostic tests as one of our limitation in the discussion line 334-336.

Reviewer 3, Comment 7: Line 101: Is the annual medical mandatory? Is there potential for selection bias?

Response: The annual medical is mandatory and is a wellness check done to ascertain miners’ overall health. By approaching potential participants at occupational health centre we ensured that all miners would be given an opportunity to be screened for eligibility for the study. 

Reviewer 3, Comment 8: Line 109: how was silicosis determined?

Response: Silicosis was determined by reviewing patient records based on the most recent CXR taken as part of routine annual examination at the Occupational Health Centre to confirm if there was evidence of silicosis, current TB and prior TB. When a potential participant was identified during pre-screening, their CXR and CXR report were requested from the radiographer and the research nurse reviewed the CXR report to verify if there was evidence of silicosis was documented. We have now made this clear in the methods section line 132-134

Reviewer 3, Comment 9: Line 134: The sample size calculated with 769 yet the final sample was only 349. There is no discussion on this as a possible study limitation in the Discussion.

Response: Thank you for the comment. We acknowledge that we did not meet the estimated sample size of 769. We have now included the following sentence in the discussion line 352-353.

“Unfortunately our target sample size was not reached resulting in limited adjusted analyses and reduced power to detect risk factors for TB infection”.

Reviewer 3: Comment 10: Line 145/6: Is this for all TB results at 0, 4 and 12 months?

Response: The main outcome was proportion of participants who were TB uninfected, defined as being negative on QFT-Plus alone. To maximize our sample size for this outcome we used the test done at enrolment for participants enrolled under either version of the protocol. For the stricter definition of being TB infected, we used the TST and QFT-plus done within 7 days of each other, either at 90 days post enrolment or at enrolment for participants enrolled under the 1st or 2nd version of the protocol, respectively. We have made changes to the methods to clarify this (line 175-185). 

In addition participants with a negative QFT result at enrolment, were followed-up 12 months post enrolment for a further QFT and TST. See response to reviewer3, comment 1. We now describe this more clearly in the methods (line 165-167) and report the data (albeit limited) in the results (237-241).

Reviewer 3, Comment 11: Line 157: ditto

Response: Please also see our clarification to reviewer 3, comment 10. For the resister definition we used QFT and TST measurements taken within 7 days of each other, either at 90 days post enrolment (initial version of the protocol) or at enrolment (subsequent version of the protocol ). We have made changes to the methods to clarify this. 

Reviewer 3, Comment 12: Line 170: why was no adjustment done on the restricted analysis?

Response: For results presented in Table 2 (defined as TB uninfected based on QFT-plus results), we had not shown an adjusted analysis for the group restricted to Black/African miner. We have now included that adjusted analysis showing a similar model as with “all miners”: Table 2a shows unadjusted and adjusted analysis among “all miners” (n=307) and 2b showing similar analyses restricted to “Black/African miners” (n=281).

For results presented in Table 3 (TB uninfected based on a negative QFT-plus/TST=0mm), only an unadjusted analysis was conducted. As stated in the results section line 247-248, we decided not conduct a multivariable analysis as the number of outcomes was small. For “all miners” – there are only 35 outcomes (out of n=136) and for “Black/African” miners there are only 23 outcomes (out of n=123).

Reviewer 3, Comment 13: Line 245: see the Major Comment number 3.

Response: We have now addressed the limitation of existing tests of infection as they relate to humoral versus adaptive responses in discussion section line 315-335. See also our response to Major Comment 3.

Reviewer 3, Comment 14: Table 2: Should the restricted group also have been adjusted?

Response: As stated above we have now added an adjusted analysis for the restricted group of miners of Black/African ethnicity (n=281), showing a similar model as with “all miners”. 

Reviewer 3, Comment 15: Table 2: Would state the factors adjusted for in the footnote of the table.

Response: Thank you for the comment. Table 2 has now been split into table 2a and 2b. In table 2a we show the unadjusted and adjusted analysis among “all miners” and in 2b a similar analysis restricted to “Black/African miners”. We adjusted for age (a priori), living quarters (live), occupation and BMI and this is now stated this in the footnote.

Reviewer 3, Comment 16: Line 344: a few grammar errors in the references. If using abbreviations for journals rather use this throughout.

Response: We have rechecked the references for grammar errors and to confirm consistency with PLOS requirements. 

Reviewer 3, Comment 17: There is no discussion on the boosting effect and if this could have been a source of bias or confounding.

Response: As mentioned above, under the first version of the protocol we collected a blood sample for QFT at enrolment followed by another blood sample for QFT and TST placed 90 days after enrolment. This was changed later to simplify the testing such that both QFT and TST were done at enrolment. We therefore think that the main analyses presented (tables 2a, 2b and 3) do not suffer from a boosting effect.

Reviewer 3, Comment 18: There is no discussion on whether silicosis could have been acting as a confounder. There was mention that miners were excluded from the study if found to have silicosis but how was the diagnosis of silicosis determined? Could resistors have had much lower exposure to silica dust by virtue of their occupation?

Response: As discussed above silicosis was determined by reviewing the most recent CXR and CXR report that were taken as part of the annual medical examination. In our study we excluded miners with silicosis based on the most recent CXR taken as part of routine care.

Reviewer 3, Comment 19: There are no references to Figure 1 and 2 in the text.

Response: Figure 1 which gives a summary of participant flow was referred in the Results section headed “Participants” line 195 and 196 and subsequently in the results section headed “Epidemiological factors associated with being TB uninfected” line 236.

Figure 2 which shows the distribution of TST among the responses among participants with both QFT and TST results available (n=235) is referred to in the results section, subsection headed “Quantiferon results and TST distribution”.

---

## [Decision Letter · Decision Letter 1]

25 Nov 2021

PONE-D-21-05516R1Resistance to Mycobacterium tuberculosis infection among  highly TB exposed South African gold minersPLOS ONE

Dear Dr. Chihota,

Thank you for submitting your manuscript to PLOS ONE. After careful consideration, we feel that it has merit but does not fully meet PLOS ONE’s publication criteria as it currently stands. Therefore, we invite you to submit a revised version of the manuscript that addresses the points raised during the review process.

Please submit your revised manuscript. If you will need significantly more time than this to complete your revisions, please reply to this message or contact the journal office at plosone@plos.org. Please include the following items when submitting your revised manuscript:A rebuttal letter that responds to each point raised by the academic editor and reviewer(s). You should upload this letter as a separate file labeled 'Response to Reviewers'.A marked-up copy of your manuscript that highlights changes made to the original version. You should upload this as a separate file labeled 'Revised Manuscript with Track Changes'.An unmarked version of your revised paper without tracked changes. You should upload this as a separate file labeled 'Manuscript'.If applicable, we recommend that you deposit your laboratory protocols in protocols.io to enhance the reproducibility of your results. Protocols.io assigns your protocol its own identifier (DOI) so that it can be cited independently in the future. For instructions see: https://journals.plos.org/plosone/s/submission-guidelines#loc-laboratory-protocols. Additionally, PLOS ONE offers an option for publishing peer-reviewed Lab Protocol articles, which describe protocols hosted on protocols.io. Read more information on sharing protocols at https://plos.org/protocols?utm_medium=editorial-email&utm_source=authorletters&utm_campaign=protocols.

We look forward to receiving your revised manuscript.

Kind regards,

Frederick Quinn

Academic Editor

PLOS ONE

Journal Requirements:

Reviewers' comments:

Reviewer's Responses to Questions

**Comments to the Author**

1. If the authors have adequately addressed your comments raised in a previous round of review and you feel that this manuscript is now acceptable for publication, you may indicate that here to bypass the “Comments to the Author” section, enter your conflict of interest statement in the “Confidential to Editor” section, and submit your "Accept" recommendation.

Reviewer #1: (No Response)

Reviewer #3: All comments have been addressed

2. Is the manuscript technically sound, and do the data support the conclusions?

Reviewer #1: Yes

Reviewer #3: Yes

3. Has the statistical analysis been performed appropriately and rigorously? 

Reviewer #1: Yes

Reviewer #3: Yes

4. Have the authors made all data underlying the findings in their manuscript fully available?

Reviewer #1: Yes

Reviewer #3: Yes

5. Is the manuscript presented in an intelligible fashion and written in standard English?

Reviewer #1: Yes

Reviewer #3: Yes

6. Review Comments to the Author

Reviewer #1: The authors have responded appropriately to my comments and the manuscript reads well. However, I do have few comments on this version which I hope will be of value.

Substantive

1. It would be worth explaining to readers why the study excludes miners “at higher risk of TB disease”. Intuitively, one would think that these states would yield some information about resistance. (Also “chronic medication” is vague).

2. The hostel exposure is a relevant hypothesis, but I don’t see support for it and don’t believe the finding merits inclusion in the Abstract. The adjusted value in Table 2 has a very wide confidence interval. The low p-value in Table 3 (unadjusted) is presumably driven by the high OR for “other mining accommodation” and is therefore not very informative.

3. The inability to adjust for confounding in the sensitivity analysis should be included under Limitations.

Minor

Abstract. The last sentence should be moved up to Results. That leaves some space to expand on the Conclusions.

P. 3, lines 72-73: Sentence unclear.

P, 4, line 77: Suggest “..general population. However, resistance..”.

P. 4, lines 83-85: Suggest “..of TB exposure status. The study was not designed to be restricted to …. nor did it specifically characterise”.

P. 6, line 137: Suggest “..whereby an individual who did not satisfy an earlier stage was excluded..”.

P. 6, last sentence: Difficult to follow. Needs rewriting.

P. 8, line 177: Suggest: “For purposes of a sensitivity analysis using a stricter definition of infection, we included participants..”. Next sentence: “The stricter definition……………was that of being QFT negative…” .

P. 9, line 200: “To increase the likelihood of identifying epidemiologic factors..” would be a more accurate lead-in on the grounds of dilution of exposure (as per lines 185-186). However, the loss of power rather undermines this goal.

P. 11, lines 262-263. This sentence duplicates the sentence at line 258.

***

Reviewer #3: Thank you for attending to the comments and adjusting the manuscript accordingly. I think the reviewer comments have been adequately taken into account and the methods and results sections are now clear. The paper provides important signals that there seems to be a resistor phenotype that mitigates the risk of TB infection.

7. PLOS authors have the option to publish the peer review history of their article (what does this mean?). If published, this will include your full peer review and any attached files.

Reviewer #1: No

Reviewer #3: **Yes: **Dr Dave Knight

---

## [Author Response · Author response to Decision Letter 1]

31 Dec 2021

PLOS ONE Editorial Office

1265 Battery Street

San Francisco, CA 94111

United States

30 December 2021

Dear Sir/Madam,

RE: Submission of revised manuscript (PONE-D-21-05516R1): Resistance to Mycobacterium tuberculosis infection among highly TB exposed South African gold miners

On behalf of my co-authors, I would like to thank the reviewers for further careful and constructive comments of the above named manuscript. The comments have helped us in further revising our manuscript.

We have addressed each comment and where applicable have made changes in the revised manuscript. All changes made in the manuscript have been spelt out in each of the responses to the comments. The reviewers’ comments are italicised and our responses are given after each comment and are in normal font. We have not made any changes to the reference list. There are no changes to the financial disclosure. 

The manuscript had been checked to confirm that it meets the PLOS ONE requirements and we have also provided details of the repository where the data are provided.

Sincerely

Violet Chihota

Chief Specialist Scientist

Aurum Institute

 

Responses to Editor and reviewer comments

PONE-D-21-05516R1

Resistance to Mycobacterium tuberculosis infection among highly TB exposed South African gold miners

Journal Requirements:

Response: We have reviewed the reference list and can confirm that it is complete and correct. 

Reviewers' comments: 

Reviewer's Responses to Questions

Comments to the Author

1. If the authors have adequately addressed your comments raised in a previous round of review and you feel that this manuscript is now acceptable for publication, you may indicate that here to bypass the “Comments to the Author” section, enter your conflict of interest statement in the “Confidential to Editor” section, and submit your "Accept" recommendation.

Reviewer #1: (No Response)

Reviewer #3: All comments have been addressed

Response: Thank you for the feedback________________________________________

2. Is the manuscript technically sound, and do the data support the conclusions?

Reviewer #1: Yes

Reviewer #3: Yes

Response: Thank you for the feedback________________________________________

3. Has the statistical analysis been performed appropriately and rigorously? 

Reviewer #1: Yes

Reviewer #3: Yes

Response: Thank you for the feedback________________________________________

4. Have the authors made all data underlying the findings in their manuscript fully available?

Reviewer #1: Yes

Reviewer #3: Yes

Response: The data are provided on the London School of Hygiene and Tropical Medicine (LSHTM) Data Compass repository through request – please see https://doi.org/10.17037/DATA.00002424.

• The study was conducted prior to data sharing being the norm and the participant information sheet was silent on anonymized data being shared, with or without restrictions, on data repositories 

• For a previous publication for the same study (Ntshiqa T et al. Gates Open Research 2021) we sought advice from the University of Witwatersrand Human Research Ethics Committee regarding data sharing, as we were also requested by the journal to share data with no restrictions. The University of Witwatersrand Human Research Ethics Committee gave us approval to share an anonymized dataset but advised that option to request permission to access the data via LSHTM data compass to be used. 

• We therefore seek approval for the data to be shared in accordance with ethics approval provided by the University of Witwatersrand Human Research Ethics Committee (Ethics Reference 15027).

5. Is the manuscript presented in an intelligible fashion and written in Standard English?

Reviewer #1: Yes

Reviewer #3: Yes

Response: Thank you for the feedback________________________________________6. Review Comments to the Author

Reviewer #1: The authors have responded appropriately to my comments and the manuscript reads well. However, I do have few comments on this version which I hope will be of value.

Response: Thank you for the feedback. We have provided point by point responses 

 below.

Substantive comments

1. It would be worth explaining to readers why the study excludes miners “at higher risk of TB disease”. Intuitively, one would think that these states would yield some information about resistance. (Also “chronic medication” is vague).

Response: The HETU study aimed to identify miners who were highly exposed to Mycobacterium tuberculosis but resist TB infection. By studying human subjects that appear resistant to TB infection enabled us to identify the mechanisms of protection. 

When designing this study we excluded individuals with prevalent TB disease. In order not to take a sputum from all enrollees to assess for prevalent TB we conducted a pre-screen using as a marker for prevalent TB i.e those “at a higher risk of TB disease” (ie., TB preventive treatment, any serious/chronic medical condition, HIV, BMI <18.5, silicosis) as well as using their most recent CXR to exclude those with TB on CXR. Sputum was subsequently taken from a subset and those culture positive were also excluded.

As stated above we also excluded individuals with any serious/chronic medical conditions. In the manuscript we had in error described this as “chronic medication” and have now rephrased to “any serious or medical condition” on page 6 line 137-138. 

2. The hostel exposure is a relevant hypothesis, but I don’t see support for it and don’t believe the finding merits inclusion in the Abstract. The adjusted value in Table 2 has a very wide confidence interval. The low p-value in Table 3 (unadjusted) is presumably driven by the high OR for “other mining accommodation” and is therefore not very informative.

Response: Thank you for the comments. We agree that in table 3 that P=0.04 reflects an overall P-value for “live” variable and is being driven by “other mining accommodation” odds ratio. 

In the abstract page 2 line 45, we have deleted the following sentence “……..with miners living in mine hostels having a lower odds of being a resister (OR 0.86; 95% CI 0.24-3.03)”. 

On page 12, line 284, we have also deleted the following sentence “In our study, the association between being TB uninfected and living condition (hostel vs non-hostel dwelling) is consistent with findings from historic studies (21)”. 

3. The inability to adjust for confounding in the sensitivity analysis should be included under Limitations.

Response: We agree. In the sensitivity analysis, due to the small number of outcomes multivariable analysis was not done and results observed in the univariate analysis might not reflect the actual relationship. We have now added a sentence “The inability to adjust for confounding in the sensitivity analysis is a limitation and unadjusted results might not reflect the actual relationship” on page 15 line 355-357.

Minor

Abstract. The last sentence should be moved up to Results. That leaves some space to expand on the Conclusions.

Response: The sentence “No epidemiological factors for being TB uninfected were identified”, has been moved from line 49 to line 44.

In the abstract, conclusions sections (line 47), we have added the following “…….and are without distinguishing epidemiological characteristics.” 

P. 3, lines 72-73: Sentence unclear.

Response: The sentence: “Household contacts have a greater risk of Mtb infection and disease than in the community in high TB burden countries” has been deleted. 

P, 4, line 77: Suggest “..general population. However, resistance..”.

Response: The sentence “Gold miners in South Africa have a greater risk of Mtb infection and disease, than the general population however resistance to TB infection in this population is poorly understood” has been changed to: 

“Gold miners in South Africa have a greater risk of Mtb infection and disease, than the general population. However resistance to TB infection in this population is poorly understood”. See lines 78-79.

P. 4, lines 83-85: Suggest “..of TB exposure status. The study was not designed to be restricted to …. nor did it specifically characterise”.

Response: Thank you for the suggestion. 

The sentence “Participants were included in the study irrespective of TB exposure status and the study was not designed to restrict to individuals who were highly exposed to TB and did not specifically characterise the resister phenotype with respect to TST and IGRA” has been changed to

“Participants were included in the study irrespective of TB exposure status. The study was not designed to restrict to individuals who were highly exposed to TB nor did it specifically characterise the resister phenotype with respect to TST and IGRA” On page 4 line 85-87

P. 6, line 137: Suggest “..whereby an individual who did not satisfy an earlier stage was excluded..”.

Response: Thank you for the suggestion.

The sentence “The full screen was done in five stages (Fig 1), whereby if an individual did not satisfy an earlier stage they were excluded immediately” has been changed to:

“The full screen was done in five stages (Fig 1), whereby an individual who did not satisfy an earlier stage was excluded.” 

P. 6, last sentence: Difficult to follow. Needs rewriting.

Response: We agree, the last sentence on page 6 (lines 143-147) “Of those satisfying both screening phases (high cumulative exposure to Mtb based on proxies of age and years in the workforce, no TB disease or not being at higher risk of TB disease and agreed to have blood draw), we then assessed for TB infection using TST and QuantiFERON-TB Gold Plus (QFT-Plus) to identify individuals who resist TB infection” is long and difficult to follow. We have changed it to:

“Of those satisfying both screening phases (high cumulative exposure to Mtb based on proxies of age and years in the workforce; no TB disease; or not being at higher risk of TB disease), we then assessed for TB infection using TST and QuantiFERON-TB Gold Plus (QFT-Plus).

P. 8, line 177: Suggest: “For purposes of a sensitivity analysis using a stricter definition of infection, we included participants..”. Next sentence: “The stricter definition……………was that of being QFT negative…” .

Response: In line 174-178, the sentence ‘’In a sensitivity analysis we included participants with QFT-Plus and TST measurements within seven days of each other: either 90 days after enrolment for participants enrolled under the first version of the protocol or at baseline for participants enrolled under the second version of the protocol. A stricter definition for being TB uninfected (subsequently referred to as being a resister), using QFT-Plus and TST, was defined as being QFT negative and having a zero TST response (TST=0mm), and TB infected as QFT-Plus positive and TST positive (induration ≥5mm)” has been rephrased to:

For purposes of a sensitivity analysis using a stricter definition, we included participants with QFT-Plus and TST measurements within seven days of each other: either 90 days after enrolment for participants enrolled under the first version of the protocol or at baseline for participants enrolled under the second version of the protocol. The stricter definition for being TB uninfected (subsequently referred to as being a resister), was being QFT negative and having a zero TST response (TST=0mm), and TB infected as QFT-Plus positive and TST positive (induration ≥5mm).

P. 9, line 200: “To increase the likelihood of identifying epidemiologic factors..” would be a more accurate lead-in on the grounds of dilution of exposure (as per lines 185-186). However, the loss of power rather undermines this goal.

Response: We have rephrased the sentence, “A further analysis was done, restricted to participants of Black/African ethnicity who, in this setting are more likely to have high cumulative exposure to Mtb. To identify epidemiological factors associated with being a resister, a sensitivity analysis using the stricter definition of being TB uninfected and restricted to participants of Black/African ethnicity was conducted” to: 

“To increase the likelihood of identifying epidemiologic factors a further analysis was done, restricted to participants of Black/African ethnicity who, in this setting are more likely to have high cumulative exposure to Mtb. To identify epidemiological factors associated with being a resister, a sensitivity analysis using the stricter definition of being TB uninfected and restricted to participants of Black/African ethnicity was conducted (Line 195-196).

P. 11, lines 262-263. This sentence duplicates the sentence at line 258.

Response: In line 262-263, the sentence referring to “…..multivariable analysis was not conducted…” is not a duplicate and is for the study sample n=136. Whereas the last sentence in line 258 refers to sample n=123. 

We have now made it more clear by rephrasing the sentence to “Again multivariable analysis was not conducted, due to the small number of outcomes” (Line 259)

The sentence “…..multivariable analysis was not conducted…..” was duplicated in line 257 and has been deleted.

***

Reviewer #3: Thank you for attending to the comments and adjusting the manuscript accordingly. I think the reviewer comments have been adequately taken into account and the methods and results sections are now clear. The paper provides important signals that there seems to be a resistor phenotype that mitigates the risk of TB infection.

Response: Thank you for the comments. 

7. PLOS authors have the option to publish the peer review history of their article (what does this mean?). If published, this will include your full peer review and any attached files.

Do you want your identity to be public for this peer review? For information about this choice, including consent withdrawal, please see our Privacy Policy.

Reviewer #1: No

Reviewer #3: Yes: Dr Dave Knight

---

## [Decision Letter · Decision Letter 2]

23 Feb 2022

Resistance to Mycobacterium tuberculosis infection among highly TB exposed South African gold miners

PONE-D-21-05516R2

Dear Dr. Chihota,

We’re pleased to inform you that your manuscript has been judged scientifically suitable for publication and will be formally accepted for publication once it meets all outstanding technical requirements.

Kind regards,

Frederick Quinn

Academic Editor

PLOS ONE

Additional Editor Comments (optional):

Reviewers' comments:

Reviewer's Responses to Questions

**Comments to the Author**

1. If the authors have adequately addressed your comments raised in a previous round of review and you feel that this manuscript is now acceptable for publication, you may indicate that here to bypass the “Comments to the Author” section, enter your conflict of interest statement in the “Confidential to Editor” section, and submit your "Accept" recommendation.

Reviewer #3: All comments have been addressed

2. Is the manuscript technically sound, and do the data support the conclusions?

Reviewer #3: Yes

3. Has the statistical analysis been performed appropriately and rigorously? 

Reviewer #3: Yes

4. Have the authors made all data underlying the findings in their manuscript fully available?

Reviewer #3: Yes

5. Is the manuscript presented in an intelligible fashion and written in standard English?

Reviewer #3: Yes

6. Review Comments to the Author

Reviewer #3: Although the study has quite a few limitations and cannot adequately adjust for confounding it still presents an important piece of work on TB. Identifying highly exposed individuals who do not seem to become infected with TB points to additional research needed to explain this. The comments seem to have been addressed. The data supports the conclusions. The statistical approach is sound. The data is available and the manuscript is now well written.

7. PLOS authors have the option to publish the peer review history of their article (what does this mean?). If published, this will include your full peer review and any attached files.

Reviewer #3: No

---

## [Editor Report · Acceptance letter]

10 Mar 2022

PONE-D-21-05516R2 

Resistance to *Mycobacterium tuberculosis* infection among highly TB exposed South African gold miners 

Dear Dr. Chihota:

I'm pleased to inform you that your manuscript has been deemed suitable for publication in PLOS ONE. Congratulations! Your manuscript is now with our production department. 

Kind regards, 

on behalf of

Dr. Frederick Quinn 

Academic Editor

PLOS ONE